# Role of NLRP7 in Normal and Malignant Trophoblast Cells

**DOI:** 10.3390/biomedicines10020252

**Published:** 2022-01-24

**Authors:** Roland Abi Nahed, Maya Elkhoury Mikhael, Deborah Reynaud, Constance Collet, Nicolas Lemaitre, Thierry Michy, Pascale Hoffmann, Frederic Sergent, Christel Marquette, Padma Murthi, Tiphaine Raia-Barjat, Nadia Alfaidy, Mohamed Benharouga

**Affiliations:** 1Institut National de la Santé et de la Recherche Médicale U1292, Biologie et Biotechnologie pour la Santé, 38054 Grenoble, France; rolandabinahed@gmail.com (R.A.N.); mayakhoury97@gmail.com (M.E.M.); deborah.reynaud.89@gmail.com (D.R.); constance.collet@cea.fr (C.C.); nicolas.lemaitre@cea.fr (N.L.); tmichy@chu-grenoble.fr (T.M.); phoffmann@chu-grenoble.fr (P.H.); frederic.sergent@cea.fr (F.S.); christel.marquette@cea.fr (C.M.); 2Commissariat à l’Energie Atomique et aux Energies Alternatives (CEA), Biosciences and Biotechnology Institute of Grenoble, 38054 Grenoble, France; 3Service Obstétrique & Gynécologie, Centre Hospitalo-Universitaire Grenoble Alpes, University Grenoble-Alpes, CEDEX 9, 38043 Grenoble, France; 4Monash Biomedicine Discovery Institute, Monash University, Clayton, VIC 3168, Australia; padma.murthi@monash.edu; 5Department of Obstetrics and Gynecology, University of Melbourne, Parkville, VIC 3010, Australia; 6Department of Gynecology and Obstetrics, University Hospital, 42100 Saint Etienne, France; tiphaine.barjat@chu-st-etienne.fr

**Keywords:** NLRP7, inflammasome, choriocarcinoma, pregnancy, maternal immune tolerance, tumor microenvironment

## Abstract

Gestational choriocarcinoma (CC) is an aggressive cancer that develops upon the occurrence of abnormal pregnancies such as Hydatidiform moles (HMs) or upon non-molar pregnancies. CC cells often metastasize in multiple organs and can cause maternal death. Recent studies have established an association between recurrent HMs and mutations in the *Nlrp7* gene. NLRP7 is a member of a new family of proteins that contributes to innate immune processes. Depending on its level of expression, NLRP7 can function in an inflammasome-dependent or independent pathway. To date, the role of NLRP7 in normal and in malignant human placentation remains to be elucidated. We have recently demonstrated that NLRP7 is overexpressed in CC trophoblast cells and may contribute to their acquisition of immune tolerance via the regulation of key immune tolerance-associated factors, namely HLA family, βCG and PD-L1. We have also demonstrated that NLRP7 increases trophoblast proliferation and decreases their differentiation, both in normal and tumor conditions. Actual findings suggest that NLRP7 expression may ensure a strong tolerance of the trophoblast by the maternal immune system during normal pregnancy and may directly affect the behavior and aggressiveness of malignant trophoblast cells. The proposed review summarizes recent advances in the understanding of the significance of NLRP7 overexpression in CC and discusses its multifaceted roles, including its function in an inflammasome-dependent or independent pathways.

## 1. Introduction

The placenta is a highly specialized organ that develops during pregnancy to ensure the growth and the normal progress of the pregnancy. This organ also plays an important role in protecting the fetus from environmental harms, including those emanating from the maternal immune system [1]. Among important cells that form the placenta is the cytotrophoblast, a cell type capable of strong adaptation to its environment, as it expresses both the maternal and paternal antigens. Numerous studies have described the trophoblast as a special cell that possesses means to prevent its recognition by the maternal immune system [2].

During early pregnancy, cytotrophoblast cells (CT) differentiate into two main cell types: the syncytiotrophoblast (ST), which represents the endocrine unit of the placenta, as it is responsible of the production and secretion of the key hormone human chorionic gonadotropin (βCG), and the second cell type is the extravillous trophoblast (EVT). EVTs are strong remodelers of the maternal spiral arteries during the first trimester of pregnancy, as they contribute to the establishment of the fetomaternal circulation [3]. As they migrate, EVTs acquire a new repertoire of proteins that belongs to endothelial cells [4]. At the port of the chorionic villi, attached proliferating cytotrophoblasts express adhesion molecules characteristic of epithelial cells such as integrins α6/β1 and αv/β5. As these cells enter the invasive cell columns, they lose the expression of epithelial cell-like adhesion molecules and acquire the expression of endothelial cell adhesion markers such as integrins α1/β1, αv/β3, and VE-cadherin, which promote vascular mimicry [5,6]. Importantly, this switch allows the heterotypic adhesive interactions that allow fetal and maternal cells to cohabit in the uterine vasculature during normal pregnancy [4].

Due to the proliferative, migratory and invasive characteristics of EVT, the initial low capacitance/high resistance of the uterine arteries is converted into high capacitance/low resistance vessels. Upon this invasion, the fetomaternal circulation is established, allowing for an increase in oxygen pressure within the intervillous space, from 20 mmHg in the early first trimester to 55 mmHg in the late first trimester of pregnancy [7].

Failure in the process of invasion of trophoblast cells into the maternal decidua has been reported to cause deregulations in the production of multiple factors, including inflammatory cytokines, reactive oxygen species (ROS) and other harmful molecules, such as uric acid [8]. These molecules have been reported to cause cellular damages and to activate intracellular processes such as inflammasomes [9]. Several have demonstrated that superficial invasion of EVT is observed in the pathology of preeclampsia and fetal growth restriction (FGR) [10,11,12,13,14], while excessive invasion of the maternal decidua and myometrium by these cells is observed in gestational trophoblastic diseases (GTD) [15,16]. GTDs are a rare subset of placental conditions encompassing benign proliferations called partial (PHM) or complete hydatidiform moles (CHM), and their invasive counterpart named gestational trophoblastic neoplasia (GTN), of which choriocarcinoma (CC) is the most aggressive [17,18,19].

Currently, increasing literature reports the involvement of inflammasomes in key processes of placental development, including trophoblast invasion [20]. Importantly, a compelling link between pregnancy pathologies, in particular GTDs, and the occurrence of mutations in one of the inflammasome genes, *nlrp7*, is now established [16,21], suggesting its involvement in the etiology of these pathologies, especially those associated with trophoblastic gestational neoplasia. Nevertheless, the underlying molecular mechanisms are largely unknown.

## 2. NLRP7 Inflammasome: Generalities

### NLRP7 Expression Pattern and Functions

Inflammasomes are cytosolic multi-protein complexes that link pathogen recognition by specific cytosolic pattern recognition receptors (PRRs) [22,23]. The nucleotide-binding and oligomerization domain (NOD)-like receptor (NLR) serves as intracellular guards that coordinate the innate immunity and inflammatory responses upon the perception of adverse signals within the cell [22,23]. The activation of these inflammasomes is mediated by two signals. The first signal activates the nuclear factor (NF-κB) pathway that induces the transcription of the pro-IL-1β and pro-IL-18 [24], while the second signal is the direct sensing of the stimulus by the inflammasome, mediating its assembly and later the maturation of the pro-IL-1β and pro-IL-18 in a caspase-1 dependent manner (Figure 1).

Among the most studied NLRs are the NLRP family members. These inflammasomes are formed from three domains: PYRIN, nucleotide binding domain (NATCH) and leucine-rich repeats (LRR), the latter of which binds to NATCH in the inactive state of the complex (Figure 1). Upon activation, LRR dissociates from NATCH-PYRIN, and then Pyrin binds to the apoptosis-associated speck-like protein (ASC) adaptor protein, which enables its subsequent binding to caspase-1, mediating the maturation of pro-IL-1β (Figure 1) and pro-IL-18. The NLRP family contains 14 members. The most studied one is the NLRP3, also reported to be associated with oncogenic functions, as it was recently reported that resistance to metastasis in *Nlrp3*^−/−^ mice was fully attributed to enhanced NK-cell activity [25]. In addition, other members such as NLRP6 have also been shown to exhibit an anti-tumor role through the suppression of the expression of pro-inflammatory cytokines in the tumor microenvironment [26].

NLRP7, also called NALP7 or PYPAF3, is an ASC-dependent inflammasome. Its gene has been reported to emerge from *NLRP2* gene [27]. Similar to all inflammasomes, NLRP7 contributes to both pro- and an anti-inflammatory processes, depending on whether it functions in an inflammasome dependent or independent pathway [23]. The function of NLRP7 inflammasome has also been reported to depend on the master regulatory transcription factor protein that controls cellular inflammation, the NF-kB [23,28]. In addition to its pro- and anti-inflammatory actions, NLRP7 plays a role in restricting intracellular bacterial replication [29] and can also induce inflammasome assembly upon specific stimulation by FSL-1 (diacylated lipoprotein) [20].

Under physiological conditions, NLRP7 mediates the maturation and secretion of IL-1β via its inflammasome’s activity [20]. However, its overexpression under pathological conditions causes an inhibition of the procaspase-1 and pro-IL-1β maturation through direct physical interaction with these pro-proteins and without any interference with the NF-kB pathway. Importantly, data from Kinoshita et al. showed that NLRP7 might exert a negative feedback loop on the transcription of IL-1β to avoid cell toxicity [9,30].

The proposed review will summarize recent advances in the understanding of NLRP7 involvement in normal and tumor placenta development. In particular, this review will report NLRP7 involvement in the development and progression of CC and discuss its multifaceted roles in the control of pregnancy inflammatory processes. The review will advance our knowledge on the underlying mechanisms of NLRP7 in the control of trophoblast differentiation and acquisition of maternal immune tolerance during pregnancy. In addition, this review will report recent advances on the mechanism by which NLRP7 contributes to the development of HM and their progression to CC.

## 3. NLRP7 and Normal Pregnancy

Beside its contribution to inflammatory responses in numerous physiological systems, NLRP7 exhibits an important role in the control of the female reproduction processes [20,28,29,31,32,33,34]. NLRP7 has been reported to be highly expressed in the oocyte and to interact with other maternal-effect genes to regulate ovarian reproductive activities [35]. Immunohistochemical localization of NLRP7 within adult ovary sections revealed that the protein was present in follicles regardless of their developmental stages [35]. In addition, NLRP7 knockdown is unfavorable for the pre-implantation embryo development, in vitro [36,37].

Because maintenance of human pregnancy is considered as an immunological paradox, it has been established that its normal outcome depends on finely tuned adaptations at the fetomaternal interface of numerous systems, including the innate and adaptive immune systems. At this interface, two distinct genomes must interact in order to maintain tolerance of the allograft and to preserve the pregnancy. In this way, the placenta has to employ several mechanisms to regulate immune tolerance and modulate the way the maternal immune system adapts in the presence of potentially dangerous signals [38,39].

Importantly, placental trophoblasts, endothelial cells and macrophages (Hofbauer cells) have been reported to be sensitive to infectious agents via the PRRs [40]. The PRRs are known to sense both pathogen-associated molecular patterns (PAMPs) and host-derived damage-associated molecular patterns (DAMPs). The latter include exosomes, reactive oxygen species (ROS), uric acid, cholesterol and microparticles [41,42]. During pregnancy, both maternal and fetal compartments have been reported to express mRNA and protein of the following NLRPs, 1 to 4 and NLRP7, as well as the adaptor protein ASC and the main caspase (caspase 1). These proteins have been reported in ST and CT in the cells of the myometrium and in the amnion cells [43,44]. At term, the activity of the inflammasomes have been reported to be increased in the cervix and the decidua [45]. In addition, recent studies have demonstrated that pyroptosis can also occur upon the activation of the NLRP3 inflammasome during the process of labor [46].

In relation to NLRP7, recent studies have demonstrated that this protein plays a central role in pregnancy-induced immune adaptations [47]. Early in pregnancy, NLRP7 is abundantly expressed in M2-polarized decidual macrophages, and its overexpression suppresses M1 and increases M2 macrophage marker expression, suggesting NLRP7 contribution to immunological homeostasis of the endometrium early in pregnancy [36]. Nevertheless, NLRP7 function as an inflammasome in early pregnancy is still elusive. While the first studies proposed an anti-inflammatory role of NLRP7 in non-immune cells, further studies have shown the assembly of a functional NLRP7 inflammasome [28]. NLRP7 has also been reported to contribute to the decidualization process. Its silencing has been reported to impair decidualization, and its NLRP7 expression enhances this process [36].

Recently, our group demonstrated that NLRP7 expression is also critical in the early stages of placental development, particularly in trophoblast cells [20]. NLRP7 increased cytotrophoblast proliferation and controlled their precocious differentiation toward EVT and ST. Importantly, these effects were dependent upon the oxygen tension, a key parameter of placental development during the first trimester of pregnancy, and by the endocrine regulation of the key hormone, βhCG [20].

High levels of NLRP7 in hypoxia were linked to a lower differentiated state from CT to ST and EVT and to a higher proliferative state. These data suggested that NLRP7 is associated with an undifferentiated state of trophoblast cells. These findings are in line with recent data published by Alici-Garipcan et al. [48], who demonstrated that NLRP7 downregulation allows the cells to better respond to BAP (a BMP4 inhibitor), allowing for their commitment toward a trophoblastic lineage. They further confirmed their results by demonstrating that one of the ESC (embryonic stem cell) genes, the OCT3/4, was downregulated upon NLRP7 downregulation. Importantly, restoring of NLRP7 expression not only allows the cells to recover from the low expression of BMP4 but also restores the expression of OCT3/4. These data strongly suggest that NLRP7 is involved in the control of trophoblast lineage and commitment. This means that NLRP7 inhibits pluripotent stem cells (iPSCs) commitment toward the trophoblast lineage through a BMP4 dependent pathway [48]. Another study using patient-specific induced pluripotent stem cells (iPSCs)-derived trophoblast cells also demonstrated that NLRP7 control of trophoblast differentiation involves members of bone morphogenetic proteins, including BMP4 [48].

## 4. NLRP7 in Pregnancy Pathologies

### 4.1. NLRP7 and Recurrent Hydatidiform Moles

Until 2014, studies on NLRP7 in relation to pregnancy pathologies have mainly been focused on its association with recurrent HM, as ample evidence has been collected to convincingly link HM to NLRP7 [9,31,49]. Several NLRP7 gene variants are clearly associated with reproduction and imprinting defects [9,31,49,50]. HM patients have been reported to carry nonsynonymous variants of NLRP7, and more than 200 sequence variants have thus far been reported in 48–80% of recurrent HM patients [50,51]. Mutations in the *nlrp7* gene include insertions, substitutions, deletions and duplications. While the association between *NLRP7* mutations and HM occurrence is convincing, the functional consequences and the underlying molecular mechanism are still unknown.

### 4.2. NLRP7 and Fetal Growth Restriction

In relation to pregnancy pathologies, we demonstrated that NLRP7 expression is elevated in the placentae of pregnancies complicated by fetal growth restriction (FGR), a pregnancy often characterized by increased inflammation [13,20,52,53]. These findings strongly suggest that the NLRP7 inflammasome could be involved in the etiology of FGR. In addition, we demonstrated that the expression of other NLRP7 inflammasome components, including ASC, cleaved caspase-1 and mature IL-1β, were also increased in FGR placentae and that circulating IL-1β, but not IL-18 levels, were significantly increased in the sera from FGR patients [20].

### 4.3. NLRP7 and Preeclampsia

While deregulation of NLRP7 expression in the placenta of preeclamptic patients is likely, assuming that inflammation is one of the main causes of PE development, no study has thus far reported NLRP7 status in relation to this significant and life-threatening pathology of pregnancy complications. Because the NLRP gene family has been reported to be associated with the etiology of imprinting defects and that PE has also been observed in disorders associated with aberrant methylation at genomically imprinted loci, it was hypothesized that the NLRP gene family may be implicated in PE. To verify this hypothesis, Soellner et al. analyzed a cohort of 47 PE patients for NLRP gene mutations using next generation sequencing [54]. The screening indicated that NLRP mutations are not a relevant cause of PE. Further studies are warranted to show the potential clinical and biological significance of NLRP7 in the etiology of PE.

Overall, the above studies describing NLRP7 expression and activation in physiological inflammation, associated or not with complicated human pregnancies such as FGR, provide important information on their potential role in the pathophysiology of pregnancy complications such GTDs.

## 5. NLRs and Cancer

Because elevated serum concentrations of IL-1β and IL-18 are often correlated to malignancies, it was suggested that all members of the NLR family are associated with pro-tumoral activities. Nevertheless, numerous studies, especially the comprehensive review by Terlizzi et al., reported that members of this family can exhibit both pro-and anti-tumoral activities, depending on the type of cancer and whether they function in inflammasome dependent or independent pathways [26].

While inflammasome activation in cancer is supposed to control its expansion, some stimuli of inflammasomes can behave as tumor promoters through the induction of chronic inflammation that rather facilitates tumor development. This sight is contrasted in animal models of colon cancer in which the activation of some inflammasome complexes is associated with tumor protection. For instance, NLRC4- and caspase-1-deficient mice have been reported to develop increased colonic inflammation, responsible for higher colon adenocarcinoma burden, in an azoxymethane/dextran sulfate sodium (AOM/DSS) mouse model. NLRC4 and caspase-1 were inferred to exert a protective function in that model via a direct effect on epithelial cell proliferation [26]. In addition, knockdown of NLRP6 in mice increased their risk of developing colorectal cancer, suggesting its significant role in the onco-suppressive activity [55]. Conversely, NLRP3, the most studied NLR has been reported to be associated with pro- and anti-carcinogenic roles. In the DSS/AOM cancer model, NLRP3 has been reported to play a protective role [56]. However, this member has been associated with poor survival rate of colorectal cancer [57] and to higher susceptibility to melanoma [58] and myeloma [59]. In addition, NLRP3 has been reported to suppress NK (natural killer) and T cell-mediated anti-tumor actions and immune-editing in a mouse model of carcinogen-induced sarcoma and metastatic melanoma [25]. This phenomenon was mediated by IL-1β-dependent recruitment of immune suppressive cells, such as myeloid-derived suppressor cells (MDSCs) and Treg cells [25]. Taken together, these findings strongly suggest that the roles of NLRs in human cancers are yet to be elucidated.

While increasing literature exists about the involvement of NLR members in cancer development, few studies are available for NLRP7. Increased NLRP7 expression has been reported to be associated with poor prognosis of colorectal cancer [60] and to play a crucial role in testicular tumorigenesis [61]. In relation to the female reproductive system, Ohno et al. demonstrated that, in endometrial cancer, a strong relationship exists between the depth of tumor myometrial invasion and NLRP7 expression [33]. The staining of NLRP7 was heterogeneous in advanced tumors and NLRP7 was frequently located at the invasion front of the tumor. The authors proposed that the expression of NLRP7 in the invasive front of cancer may provide malignant tissue with a suitable environment for their growth and spread through inducements to immunosuppression [33].

### 5.1. NLRP7 and Gestational Trophoblastic Diseases

Distinct from normal placental development, GTDs are a rare subgroup of placental pathologies, encompassing PHM or CHM and their non-molar counterpart such as CC, which constitutes the most aggressive form of placental cancer [62]. CC is a highly proliferative and invasive tumor as trophoblast cells forming the tumor metastasize into multiple organs, including the vagina, lungs and brain [18,62]. CC has an estimated incidence of 2 to 7 in 100,000 pregnancies in Europe and North America. This incidence is higher in Asia and Africa, with 5 to 202 in 100,000 pregnancies [62]. CHM is a morbid pathology that is associated with a high risk (20%) for patients to develop post-molar CC [62]. More often, CC may also develop after normal delivery. The incidence of this type of CC is 1 per 67,000 live births [19,63]. Recent studies have shown that 50% of patients with recurrent HM have mutations in the gene *nlrp7* [62]. While the association of biallelic mutations in *nlrp7* with recurrent HM is well established, its role in the development of GTDs, especially CC, is poorly understood and often controversial [28].

### 5.2. NLRP7 and Choriocarcinoma

Since the identification of *nlrp7* as a highly mutated gene in recurrent HMs, no study has been conducted to determine whether deregulations in the expression of this gene may contribute to the change in the behavior of the tumor trophoblast cells and their metastasis. Recently, we investigated the role of NLRP7 in these processes [64]. We used three approaches to define the role of NLRP7: (i) a clinical study in which we used human sera and placentae that were collected from normal pregnant women and from patients with CHM or CC; (ii) an in vitro study in which we investigated the influence of NLRP7 knockdown on the tumorigenesis of the choriocarcinoma cell line, JEG3, which used both 2D and 3D culture systems; and (iii) an in vivo study in which we used an orthotopic model of CC and a metastatic model of this cancer [65]. This study demonstrated that NLRP7 was upregulated in tumor cells, and in CHM and CC placentae. In JEG3 cells, NLRP7 increased proliferation and 3D organization of malignant cells.

NLRP7 increased expression in JEG3 cells and in CHM and CC tissues strongly suggested that its inflammasome is highly activated. Nevertheless, no production or secretion of mature IL-1β have been observed in JEG3 cells. This finding strongly suggests that NLRP7 may function in an inflammasome-independent pathway in malignant trophoblast cells. This statement is in line with previous studies reporting that overexpression of NLRP7 exerts negative feedback on the production and maturation of IL-1β. [9,30]. Recent studies from the literature demonstrated that IL-1β might negatively control the proliferation of trophoblast cells through the deregulation of the cell cycle [66,67]. Importantly, Chow et al. demonstrated that another member of the NLR family, the NLRP3, promotes metastasis in an inflammasome independent manner and that knock-out mice for NLRP3 exhibit lower numbers of lung metastases upon intravenous inoculation of prostate or melanoma malignant cells [25]. It has also been reported that overexpression of NLRP12 is associated with the aggravation of prostate cancer without any increase in the levels of mature IL-18 or IL-1β by these cells [55]. Overall, these results roughly suggest that NLRP7, similar to NLRP12 and NLRP3, functions in an inflammasome independent manner in malignant cells [55,68].

Importantly, the in vivo study that used the orthotopic model of CC, which was injected within its placenta with NLRP7 invalidated-CC cells, showed higher maternal immune response and that the mice developed smaller tumors and displayed less metastases. Furthermore, we observed a strong increase in the levels of IL-1β, both locally in mouse placenta and in the maternal serum. This finding strongly suggests that the expression of NLRP7 by the trophoblast cells contribute to its camouflage by the maternal environment (Figure 2). In line with this assumption, we observed that malignant cells that were inactivated for NLRP7 exhibited significant decrease in the expression of proteins that contribute to maternal immune tolerance. These include PD-L1, HLA-G and hCG. The latter hormone has recently been reported to increase the activity of regulatory T cells (Treg) and to retain the tolerogenic activity of dendritic cells [69,70]. Importantly, these findings strongly support a local immune tolerance that is mediated by malignant cells- secreted hCG. This hormone is known to act as a strong chemoattractant for T-suppressors that are apoptotic actors for T-lymphocytes.

From a clinical standpoint, the results obtained in vivo may elucidate what may occur in patients with CHM who go on to develop CC. Several studies have suggested that CC develops in patients with a weak immune system that facilitates a favorable environment for tumor growth [9,23]. The mouse model used in our study is an immunodeficient model that mimics the weak immune system of CHM or CC patients. These mice miss mature B and T lymphocytes, but have normal natural killer cells, macrophages and granulocytes [71].

Altogether, these findings have demonstrated that NLRP7 plays a key role in changing the behavior of trophoblast malignant cells (Figure 2). This occurs through its contribution to the establishment of an immunosuppressive maternal microenvironment that downregulates the maternal immune response and facilitates trophoblast tumorigenesis.

### 5.3. Proposed Mechanism of NLRP7 Control of Maternal Microenvironment

Because the fetus is considered as a semi-allograft by the maternal immune system, a slowdown in the function of the immune system is required [39,72]. Yet, the normal pregnancy progresses and leads to birth in most cases. This is actually due to a fine adaptation of the maternal immune system to tolerate the foreign body. Hence, one can speculate that a similar mechanism of tolerance may occur during tumor development [73]. Among proteins that contribute to trophoblast tolerance from the maternal system [39,72] are the HLA (human leukocyte antigen) family members, as well as the PDL-1/PD-1 system.

### 5.4. HLA Family in Normal and Tumor Placenta

HLA proteins are mainly expressed by the EVT. This family of proteins has been shown to be involved in the attenuation of the pool of immune cells present at the fetomaternal interface from implantation to delivery [74,75]. The HLA family is composed of numerous members that are differentially involved in the immune tolerance during pregnancy. Among all HLA members, HLA-G is exclusively expressed on EVT [76]. During pregnancy, HLA-G plays an immunosuppressive role rather than an antigen-presenting role [39,72]. In non-pathological conditions, it is expressed only at the surface of the EVT, thymic epithelial cells, the cornea and in the cells facing the amniotic fluid [74]. The main role of HLA-G is to inhibit cytotoxic T lymphocytes and Natural killer cells through an interaction with their ILT-2 and KIR receptors [77]. Recent studies also showed that HLA-G regulates trophoblast invasion, a key parameter of placental development in normal and tumor conditions [78]. In addition, soluble HLAG (sHLA-G) has been shown to impair the expression and function of different chemokines receptors in T, B, and NK cells through the ILT2 receptor [77]. HLA-G has also been shown to be upregulated by βhCG in JEG3, suggesting that this hormone also contributes to the mechanism by which choriocarcinoma cells develop immune tolerance [69,70].

### 5.5. PDL-1 in Normal and Tumor Trophoblast Cells

The survival of the trophoblast cells depend on their ability to evade the immune system through the inhibition of their anti-tumoral activity [39,72]. A common ligand found in several aggressive cells is the protein PD-L1 (programmed death ligand -1), which mediates immunosuppression upon binding to its receptor PD-1, commonly expressed by immune cells [79]. During normal pregnancy, the immunosuppressive role of PD-L1 is major, as it is expressed on the ST. PDL-1 interaction with these cells promotes an immune tolerance to the fetal tissues [80].

In CC, PD-L1 is expressed by the ST and CT. PD-L1/PD-1 interaction provides an immune tolerance through the activation of the paternal antigen-specific naïve helper T cells Tregs [81]. Importantly, we demonstrated that NLRP7 knockdown caused a decrease in PD-L1 expression, suggesting that this protein is directly involved in the NLRP7-mediated immunosuppression [64]. Altogether, these findings suggest that a tight relationship exists between the maternal immune system and the NLRP7 inflammasome. Ongoing studies are in progress to decipher the mechanisms by which this occurs at the fetomaternal interface.

## 6. Concluding Remarks

It clearly appears that appropriate NLRP7 expression and NLRP7 inflammasome activity are essential during early pregnancy. However, further investigation is required to establish how HM-associated *NLRP7* overexpression and variants might affect NLRP7 function and lead to reproductive wastage.

Overall, it appears that the NLRP7 mode of function will tightly depend on the cellular status. Under physiological conditions, NLRP7 will function in an inflammasome-dependent pathway to contribute to the maintenance of the required fine balance between pro-inflammatory and anti-inflammatory settings. This will be ensured through the processing of pro-IL-1β to IL-1β [64]. This is what clearly has been reported in normal trophoblast cells and in placental explant model systems [20]. The NLRP7 inflammasome activity can be exacerbated in the context of pregnancy pathologies such as FGR pregnancy to overcome the stressful conditions of the trophoblast cells [20].

During CC development, NLRP7 appears to function in an inflammasome-independent pathway, as no IL-1β was produced. As demonstrated by numerous studies, NLRP7 overexpression might explain its function in an inflammasome independent pathway [22,23,64]. However, the underlying mechanism is yet to be elucidated. A likely explanation drifts toward an abnormal interaction between NLRP7 protein and the NFkB pathway, leading to an inhibition of the transcription of Pro-IL-1β. The above data clearly show that the physiological expression of NLRP7 in trophoblast cells contributes to their acquisition of a normal immune tolerance and controls their precocious differentiation. In contrast, its overexpression in tumor conditions exacerbates immune tolerance, permitting these cells to proliferate and invade the maternal tissue (Figure 2).

To better illustrate the actual understandings of NLRP7 functions in normal and in malignant trophoblast cells, we designed the cartoon reported in Figure 3.

Panel A reports a normal trophoblast cell that expresses NLRP7 protein at normal levels. In this cell, NLRP7 functions in an inflammasome pathway, as it mediates the maturation of pro-IL-1β, produced upon the activation of the NF-kB pathway, into IL-1β. This maturation is due to the activation of the enzyme caspase-1. It was also supposed that NLRP7 activity in an inflammasome pathway controls the expression of HLA-G and PD-L1 [20,64], allowing trophoblast cells to further proliferate, especially during the first trimester of pregnancy. This process is driven by the hypoxic environment that dominates during early pregnancy and positively regulates NLRP7 expression [3,20,52,82].

Panel B reports a malignant trophoblast cell that overexpresses NLRP7 protein. In this cell, NLRP7 inhibits pro-IL-1β production, likely through downregulation of the NF-kB pathway. The absence of mature IL-1β in malignant cells strongly suggests that NLRP7 functions in an inflammasome-independent pathway, which contributes to the exacerbation of tumor-dependent cell proliferation and invasion. As the IL-1β free environment allows tumor cells to escape maternal immune control, these cells undergo camouflage through an increase in the expression of a selected repertoire of proteins that includes check point proteins such as PDL1; HLAG and βhCG.

In conclusion, this review summarizes the current knowledge on NLRP7 expression and function in early normal pregnancy, in FGR, and in choriocarcinoma. While a significant progress has been made to establish NLRP7 association with molar pregnancies, further investigations are required to establish its role during the early stages of pregnancies.

## Figures and Tables

**Figure 1 biomedicines-10-00252-f001:**
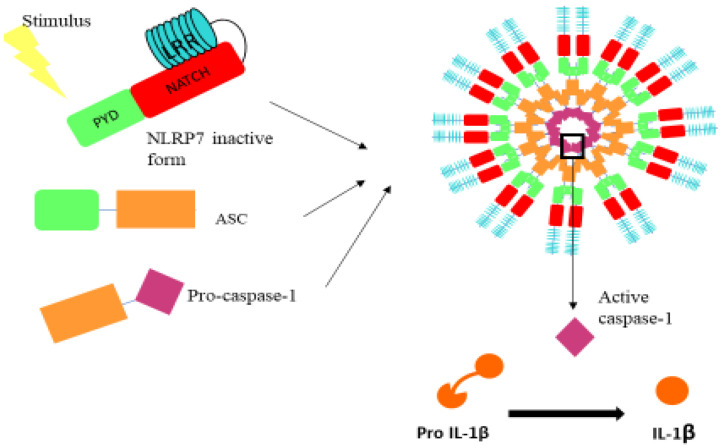
Illustration of the domains that comprise the NLRP7 inflammasome. NLRP7 inflammasome is composed of the NLRP7 receptor, the ASC adaptor protein and procaspase 1. PYD: pyrin domain; NATCH: nucleotide binding domain (allows the ATP-dependent oligomerization of the NLRs); LRR: leucine-rich repeats.

**Figure 2 biomedicines-10-00252-f002:**
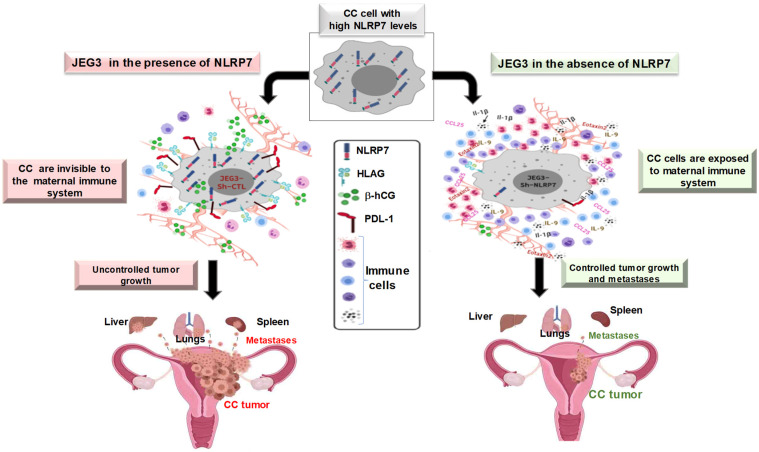
Proposed model for the role of NLRP7 protein in the development of gestational choriocarcinoma.

**Figure 3 biomedicines-10-00252-f003:**
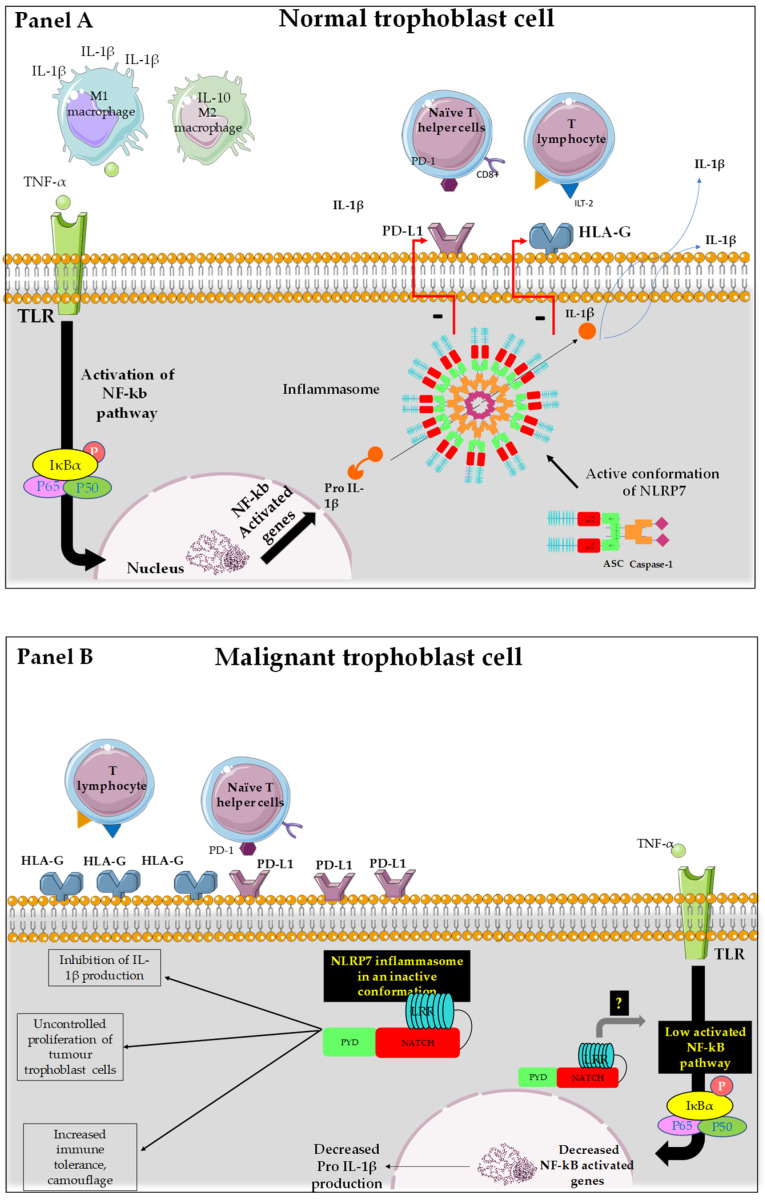
Representation of the mode of function of NLRP7 in normal and tumor trophoblast cells. (**A**): NLRP7 is normally expressed in trophoblast cells and functions in an inflammasome-dependent manner to allow maturation of pro-IL-1β to IL1β. The transcription of pro-IL-1β depends on the activation of NF-κB that translocates into the nucleus and increases the transcription of pro-IL-1β. NLRP7 also regulates the expression of HLA-G and PD-L1 to allow normal tolerance of the trophoblast by the maternal immune system and favors the polarization of macrophages to M1 subtype. All these processes allow the protection of the fetus and ensure the progress of the pregnancy. (**B**): In malignant trophoblast cells, NLRP7 is overexpressed and functions in an inflammasome-independent manner. NLRP7 in turn mediates the increase in HLA-G and PD-L1 expression. This exacerbates maternal immune tolerance and camouflage of the tumor cells, creating a favorable, ant—inflammatory environment for tumor growth. NLRP7 overexpression mediates the excessive proliferation of trophoblast cells and suppresses their differentiation, allowing for further migration and invasion that ultimately leads to metastasis of distinct maternal organs.

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
