# Peer review of "Role of NLRP7 in Normal and Malignant Trophoblast Cells"

_biomedicines, 2022, doi:10.3390/biomedicines10020252_

Round 1

Reviewer 1 Report

The review well describes recent progress on the function of NLRP7 gene towards normal and tumor trophoblasts, one for IL-1b-dependent and another for IL-1b-independent. This review article has a novelty and originality on the NLRP7 since the authors include recent research results such as those in their paper published this year (2021). I have no particular comments.

Author Response

We would like to thank this reviewer for his positive comments. The paper has been succintly improved. 

Reviewer 2 Report

This manuscript reviews the role of NLRP7 in trophoblast cells - normal  and in choriocarcinoma (CC).  It's a dense read with lots of repetition and somewhat disparate data but of interest to trophoblast biologists. 

I have many suggestions for the authors starting with the title. Perhaps instead of "tumor trophoblast cells" consider using "malignant  trophoblast cells" or something like: Role of NLRP7 in benign and malignant trophoblast. Although the only malignant trophoblast discussed in the manuscript is that of choriocarcinoma, I also would like to know if NLRP7 has been studied in the other GTN trophoblastic tumors  (epithelioid trophoblastic tumor or placental site trophoblastic tumor).  Perhaps partial hydatidiform moles should be excluded as they are essentially benign and have no significant increased post-molar choriocarcinoma risk. 

Abstract: As choriocarcinomas can follow non-molar pregnancies, even normal gestations,  the first sentence needs to be corrected.  The abstract goes on to say that this review discusses the role of NLRP7 in the "development and aggressiveness" of choriocarcinoma. I do not believe that the role in development has been discussed. Malignant transformation of benign trophoblast is not described. Only the role of NLRP7 in the "behavior" of normal and choriocarcinoma is discussed. This should be clearly stated in the abstract. I would change the sentence  on lines 39-42 to "...normal pregnancy an overexpression may contribute to choriocarcinoma's aggressive behavior".

Introduction: The first sentence should read "the growth and normal progress of the pregnancy"

Section 2: line 112 should read "NLRP family contains 14 members". Lines 123-124 should read "...protein that controls cell inflammation, NF-kB"

Section 3: and throughout...trophoblast is both plural and singular, "trophoblasts" is incorrect. One can just say "trophoblast" or "trophoblast cells". I think the authors mean inactivation instead of invalidation throughout, as in line 198.

Section 4 a. I am not  aware of  HM demonstrating chorioamniontitis or chorangiosis. In fact, complete hydatidiform moles rarely have a well developed gestational sac or villous vessels. I think this must be an error in interpretation of the data. I suggest removing this or clarifying it.

Section 4 b. Change "suffering from: to "complicated by" in line 224.

Section 4 c. "threatening pathology" on line 234 might be better said with "severe pathology" or "significant disease".

Section 5 a. Although CC can follow a CHM the data on post PHM CC is weak at best. I would state instead of PHM or CHM and their invasive counterpart CC use nonmolar counterpart. Invasive CHM is an entity too. CHM is morbid, not benign, disease and can be clinically difficult to treat. PHM do behave benignly for the most part and I think of them differently than CHM in terms of disease propensities.  The authors should state what the risk for developing CC is after a non-molar gestation.

Section 5 b The first paragraph seems out place when the second paragraph essentially negates it.  I have trouble with the assertion that the authors or the literature  shows NLRP7 plays a role in the development of CC. No such role in malignant transformation has been shown. The authors make a good argument that it plays a role in the behavior of "malignant" trophoblast from CC and that should be the focus of this section. I suggest using the present tense throughout.  Again, does "invalidated" mean "inactivated"? On page 8 line 342 suggest changing to "; yet, the normal pregnancy progresses and leads to birth in most cases".

Section 5 c Isn't it true that HLA-G is expressed only on  the EVT? Not the ST or CT?

The figures are quite complex and difficult to read. Could they be simplified? Again suggest a different term for "tumor trophoblast cells"

Author Response

Responses to the comments and Suggestions of the reviewers

We would like to thank this reviewer for his pertinent comments that will allow the improvement of the proposed review.

Q1: I have many suggestions for the authors starting with the title. Perhaps instead of "tumor trophoblast cells" consider using "malignant trophoblast cells" or something like: Role of NLRP7 in benign and malignant trophoblast. Although the only malignant trophoblast discussed in the manuscript is that of choriocarcinoma, I also would like to know if NLRP7 has been studied in the other GTN trophoblastic tumors (epithelioid trophoblastic tumor or placental site trophoblastic tumor).  Perhaps partial hydatidiform moles should be excluded as they are essentially benign and have no significant increased post-molar choriocarcinoma risk.

a-We have changed the title according to the reviewer’s suggestion

b-We have not yet considered NLRP7 in the other GTN trophoblastic tumors (epithelioid trophoblastic tumor or placental site trophoblastic tumor). This aspect is planned in future studies.

c- We agree with the reviewer that partial hydatidiform moles are essentially benign and have no significant increased post-molar choriocarcinoma risk. We have only cited PHM in this review to distinguish them for CHM. They only have been cited four times and no specific study related to this condition has been cited in the review.

Q2: Abstract: As choriocarcinomas can follow non-molar pregnancies, even normal gestations,  the first sentence needs to be corrected.  The abstract goes on to say that this review discusses the role of NLRP7 in the "development and aggressiveness" of choriocarcinoma. I do not believe that the role in development has been discussed. Malignant transformation of benign trophoblast is not described. Only the role of NLRP7 in the "behavior" of normal and choriocarcinoma is discussed. This should be clearly stated in the abstract. I would change the sentence  on lines 39-42 to "...normal pregnancy an overexpression may contribute to choriocarcinoma's aggressive behavior".

  • The abstract has been changed according to the reviewer relevant suggestions

Q3: Introduction: The first sentence should read "the growth and normal progress of the pregnancy"

  • The sentence has been changed as suggested by this reviewer.

Q4: Section 2: line 112 should read "NLRP family contains 14 members". Lines 123-124 should read "...protein that controls cell inflammation, NF-kB"

-              The sentence line 114 has been changed as suggested by this reviewer

-            The sentence lines 125-126 has been changed as suggested by this reviewer

Q5: Section 3: and throughout...trophoblast is both plural and singular, "trophoblasts" is incorrect. One can just say "trophoblast" or "trophoblast cells". I think the authors mean inactivation instead of invalidation throughout, as in line 198.

-            The word “trophoblasts” has been modified throughout the revised manuscript  and the word “inactivation or knockdown” replaced the word “invalidation”. 

Q5: Section 4 a. I am not  aware of  HM demonstrating chorioamniontitis or chorangiosis. In fact, complete hydatidiform moles rarely have a well developed gestational sac or villous vessels. I think this must be an error in interpretation of the data. I suggest removing this or clarifying it.

  • The whole sentence has been removed. We agree that the association of HM with chorioamniontitis or chorangiosis is yet to be confirmed and was only suggested in the former cited study.

Q6: Section 4 b. Change "suffering from: to "complicated by" in line 224.

  • The sentence line 220 has been changed to the one suggested by the reviewer.

Q7: Section 4 c. "threatening pathology" on line 234 might be better said with "severe pathology" or "significant disease".

We would like to keep this word to describe preeclampsia, as it is commonly used in the case of this disease. A search in Pubmed with the sentence “Preeclampsia is a life-threatening” gave at least 475 results.

Q8: Section 5 a. Although CC can follow a CHM the data on post PHM CC is weak at best. I would state instead of PHM or CHM and their invasive counterpart CC use nonmolar counterpart. Invasive CHM is an entity too. CHM is morbid, not benign, disease and can be clinically difficult to treat. PHM do behave benignly for the most part and I think of them differently than CHM in terms of disease propensities.  The authors should state what the risk for developing CC is after a non-molar gestation.

We have changed the paragraph that provides information on PHM and CHM according to the reviewer’s recommendations. Also, we have reported the incidence of non-molar CC. All these modified and inserted sentences can be found between lines 283 and 285.

Q9: Section 5 b The first paragraph seems out place when the second paragraph essentially negates it.  I have trouble with the assertion that the authors or the literature shows NLRP7 plays a role in the development of CC. No such role in malignant transformation has been shown. The authors make a good argument that it plays a role in the behavior of "malignant" trophoblast from CC and that should be the focus of this section. I suggest using the present tense throughout.  Again, does "invalidated" mean "inactivated"? On page 8 line 342 suggest changing to "; yet, the normal pregnancy progresses and leads to birth in most cases".

All changes requested by the reviewer have been included. All new adds are highlighted in red in section 5

Q10: Section 5 c Isn't it true that HLA-G is expressed only on  the EVT? Not the ST or CT?

We agree with the reviewer that HLA-G is only expressed by EVT. We have replaced the “trophoblast” denomination that includes cytotrophoblast, syncytiotrophoblast and EVT by EVT only.  We have reported trophoblast in the first version because the mRNA and protein of HLA-G are expressed in human embryos and especially on the trophectoderm, which differentiates into cytotrophoblasts and syncytiotrophoblast [1]. Also, HLAG in soluble form is also expressed in the cytotrophoblasts which undergo the syncytial fusion with the syncytiotrophoblast [2-3].

[1] A. Verloes, et al. HLA-G expression in human embryonic stem cells and preimplantation embryos. Eur. J. Immunol., 186 (2011), pp. 2663-2671

[2] J.S. Hunt. Stranger in a strange land. Immunol. Rev., 213 (2006), pp. 36-47

[3] R. Rizzo et al. The importance of HLA-G expression in embryos, trophoblast cells, and embryonic stem cells. Cell. Mol. Life Sci., 68 (2011), pp. 341-352

Q11: The figures are quite complex and difficult to read. Could they be simplified? Again suggest a different term for "tumor trophoblast cells"

As suggested by the reviewer, all figures have been simplified
